# Feasible future global scenarios for human life evaluations

Christopher Barrington-Leigh[1] & Eric Galbraith [2,3,4]

Subjective well-being surveys show large and consistent variation among countries, much of which can be predicted from a small number of social and economic proxy variables. But the degree to which these life evaluations might feasibly change over coming decades, at the global scale, has not previously been estimated. Here, we use observed historical trends in the proxy variables to constrain feasible future projections of self-reported life evaluations to the year 2050. We find that projected effects of macroeconomic variables tend to lead to modest improvements of global average life evaluations. In contrast, scenarios based on non-material variables project future global average life evaluations covering a much wider range, lying anywhere from the top 15% to the bottom 25% of present-day countries. These results highlight the critical role of non-material factors such as social supports, freedoms, and fairness in determining the future of human well-being.

[1] Institute for Health and Social Policy; and School of Environment, McGill University, Montreal H3A1A3 QC, Canada. [2] ICREA, Pg. Lluís Companys 23, 08010 Barcelona, Spain. [3] Institut de Ciència i Tecnologia Ambientals (ICTA) and Department of Mathematics, Universitat Autonoma de Barcelona, 08193 Barcelona, Spain. [4] Department of Earth and Planetary Sciences, McGill University, Montreal, QC, Canada. Correspondence and requests for materials should be addressed to C.B.-L. (email: Chris.Barrington-Leigh@McGill.ca)

C omplex policy decisions, ranging from international climate change negotiations to investments in education or infrastructure, often rely on projections of readily quantified material outcomes such as per capita income to assess the impacts on human welfare, for example, refs. [1,2]. An alternative approach, developed over the past few decades, aims to apply more direct measures of human experience that integrate many dimensions of life according to those living it. These well-being measures have the disadvantage of being subjective, and can therefore be difficult to interpret. Nevertheless, they have been shown to be consistent with external evaluations at the individual level, are reproducible over time within populations, and are increasingly embraced by decision makers as a leading objective[3–7]. They also show consistent predictive relationships with other societal variables.

Here, we focus on one measure of subjective well-being (SWB) with particularly consistent predictive relationships: the cognitive evaluation of life. Annual, near-global national samples of self-reported life evaluations, from Gallup's World Poll, have recently become available through the World Happiness Report. While the Poll's extensive questionnaire is not designed expressly for explaining differences in life quality, it does contain several questions which address dimensions of life known to be important to life evaluation. Prior work in a range of contexts has shown that, among these, income differences have significant explanatory power in accounting for variation in life evaluations, while less tangible aspects of human experience can account for as much or more of the variation[8]; see Supplementary Note 1. Following Helliwell et al.[8], we account for a portion of international differences in life evaluation using four proxy variables derived from the Gallup World Poll, which we will refer to as reflecting non-material factors (corruption[9], freedom, giving[10], and social support), and two proxy variables reflecting material factors (per capita gross domestic product (GDP) and life expectancy). When these measures are included in a simple linear model of cross-country differences, they explain roughly three-quarters of the variation among annual national averages of life evaluations, and parameters are estimated with high statistical precision, as shown by prior work[8]; see Supplementary Note 2 for details.

Static inter-country comparisons such as these have been the focus of much attention, but they do not address how the life evaluations of humans might change in the future, an important consideration for motivating and evaluating policy decisions. Although there is no mechanistic understanding on which to reliably predict these future changes directly, the time-span covered by the Gallup World Poll (2005–2016) is now sufficient to use as an empirical constraint on the feasible rates at which proxy variables might change, and reveals that strong trends have occurred in some countries (Supplementary Figure 1). The portion of life evaluations that can be predicted from the proxy variables would be expected to change accordingly, providing a means by which to estimate the range within which the future trajectories of life evaluations are most likely to fall.

Here, we use the historical survey data to develop a dynamic statistical model and evaluate feasible rates of change, and construct simple scenarios for human life evaluations in 2050, within the time frame commonly considered in forward-looking policies informed by climate model projections[11–16]. We find that significant changes in global average life evaluations are feasible, and could be either positive or negative. In addition, we find these future outcomes to have a markedly larger dependence on non-material proxy variables as compared with material ones.

## Results

**Proxy model of life evaluations**. Our statistical models predict a portion of the observed changes in life evaluations from the proxy variables using both static (across countries, XS) and dynamic (within countries comparing two periods, 2P) models (Fig. 1, see Supplementary Note 2 for details). As compared with the static XS model of differences across countries (Fig. 1a), the dynamic 2P model of within-country changes shows a relatively greater emphasis on freedom to make decisions, availability of social support, and perception of corruption. The coefficients in Fig. 1a, b are normalized to standard deviations, meaning that a 1 s.d. change in income predicts a 0.19 s.d. change of life evaluations, holding other factors constant, while a 1 s.d. change in social support predicts a 0.29 s.d. change in life evaluation. In our projections, below, we use the dynamic model (2P) to predict changes in life evaluation over several decades, though as discussed later, our main findings would be even more pronounced were the XS model coefficients to be used. We emphasize that the projections that emerge are not predictions of the future, but illustrate the range of the most feasible futures that might occur, depending on human actions.

**Projected range of global mean changes**. The range of feasible outcomes spanned by the three sets of scenarios is shown in Fig. 2. Two sets of scenarios relate to changes in material conditions. For the first of these sets (Organization for Economic Co-operation and Development (OECD) projections), both the optimistic (strong-growth) and pessimistic (weak-growth) projections show improvements of global average experienced life evaluation compared with the 2016 value of 5.24 out of 10. The optimistic case of the second material-factors pair (Material Trends) is characterized by steady growth at the 90th percentile of recently observed trends (4%/year for income and 0.55 years/year for healthy life expectancy), and this scenario's outcome agrees very closely with the optimistic OECD estimate. However, not all countries have actually experienced positive real economic growth. As a result, the corresponding pessimistic Material Trends scenario, using 10th percentile observed trends, admits the possibility of a small decline of global average life evaluations, to 5.1, by 2050.

The range of feasible outcomes encompassed by these material scenario sets is dwarfed by the range of outcomes in the non-material scenario set. In the top 10% of recent observed trends, freedom and social support grew at 2%/year and 0.6%/year, respectively, while corruption decreased at 1.3%/year. At these rates, our Non-material Trends scenario projects a radical global mean improvement of life evaluation to 6.9, which—for comparison—is close to the levels actually reported by Belgium and Costa Rica in 2016, when they had the 18th and 14th highest life evaluations[17]. Conversely, if the least favorable 10th percentile of observed non-material trends were to prevail in all countries, our projections suggest that the drop in life evaluations by 2050 would take the global average to 3.4, below the level of Egypt and India in 2016, when they ranked 118th and 120th out of 157 countries.

Figure 2 also shows our projections based on 30th and 70th percentile recent trends. These provide very similar qualitative conclusions as our primary analysis, in that the Non-material Trend scenarios encompass more extreme positive and negative possibilities than the material trends. Notably, the 30th to 70th percentile Material Trend projections span a similar range of life evaluations as that of the OECD-derived projections, although the trend-based projections are approximately 0.3 points lower than the OECD projections.

**Geography of projected changes**. We map the distribution of projected changes in life evaluation in our material OECD and Non-material Trend scenarios, according to population density

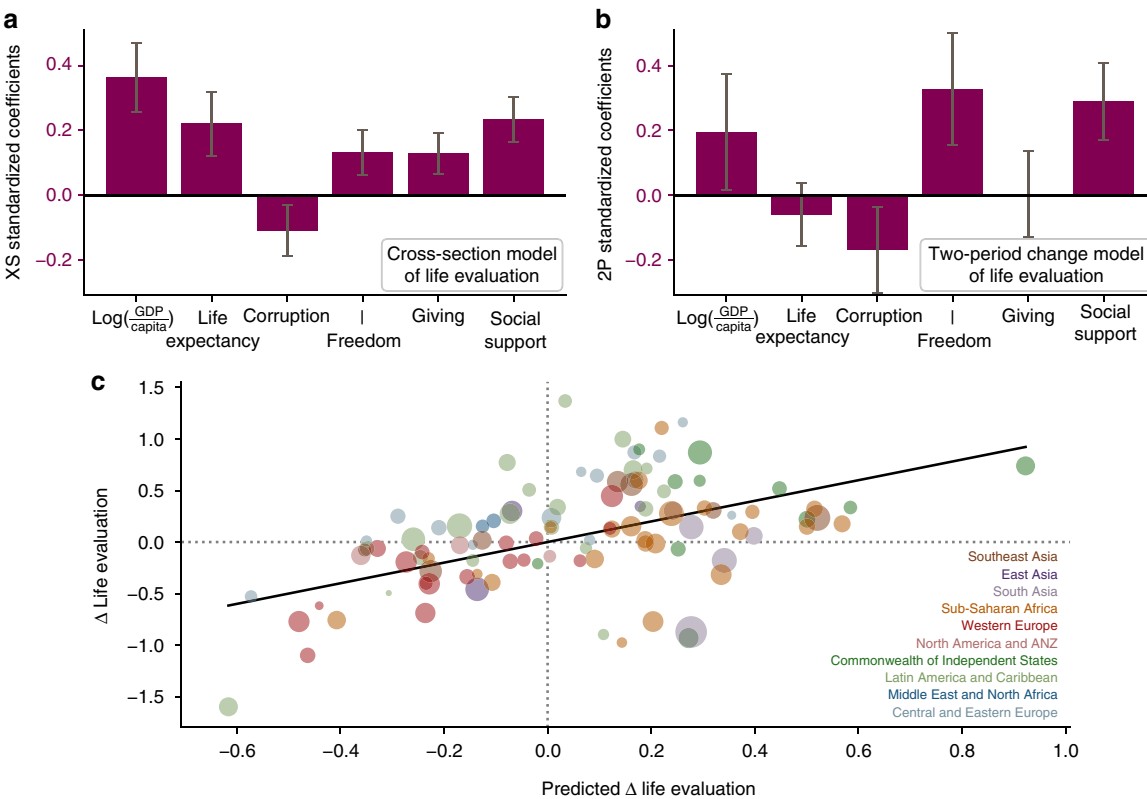

**Fig. 1** Predictions of life evaluations from proxy variables. **a** Six predictor variables are used simultaneously to predict life evaluations in static cross-section between countries (XS). All corresponding country-year observations are used, after removing global year-to-year changes. Effect sizes are shown, with 90% confidence intervals, normalized to standard deviations, in order to compare the estimated coefficients across predictor variables. The strongest predictor is the income variable, but all predictors have significant strength, and their confidence intervals largely overlap. **b** As in **a** but for the dynamic, two-period model (2P) which explains changes over time within countries. In the 2P model, confidence intervals are looser, and the income coefficient excludes zero with only ~90% confidence intervals. However, models for annual changes (see Supplementary Information) show similar patterns and have tighter confidence intervals. **c** The relationship between observed and predicted changes in life evaluation using the 2P model. The changes are the differences in national average life evaluations between 2005–2007 and 2014–2016. Symbol size corresponds to the population size of each country

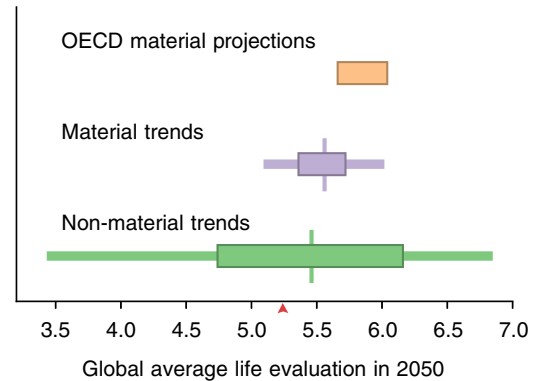

**Fig. 2** Projected feasible life evaluations in 2050 scenarios. The results of two sets of material scenarios are shown for the year 2050, based on macroeconomic projections (OECD, orange) and recent trends (purple), as well as a set of non-material scenarios based on recent trends (green). The central vertical lines indicate projections based on the median trends, the thick horizontal bar indicates the range based on the 30th to 70th %ile trends, and the thinner horizontal bar indicates the range based on the 10th to 90th %ile trends. Outcomes are calculated with coefficients from the 2P model and are weighted by projected population sizes to aggregate across countries. The red arrow indicates the population-weighted average life evaluation recorded in 2016

projections for 2050, in Fig. 3 (the corresponding Material Trend scenarios, which are uniform across countries, are shown in Supplementary Figures 2–5). For the non-material scenarios, we show only the most optimistic (90th percentile) and pessimistic (10th percentile) projections. The OECD's material scenarios are driven by national changes in per capita income, since in our 2P model life expectancy has no significant effect on life evaluations. In general, the OECD projections show some degree of improvement in all countries, even under pessimistic assumptions. Differences among countries reflect the details of the OECD's macroeconomic projections and reflect an expectation of economic convergence, that is, higher economic growth rates in countries with lower initial income.

Compared to the OECD projections, the non-material projection maps show a much wider range of possible changes, as did the global averages. In the optimistic non-material future, improvements of more than 1.5 on the 11-point life evaluation scale are widespread, including in densely populated regions of India, China, eastern Europe, and sub-Saharan Africa. The scope for improvement in non-material factors is smaller in western Europe and North America, given that these countries are already closer to the maximum values. Nonetheless, the improvements in predicted life evaluation that would appear to be feasible due to non-material changes by 2050 exceed those of even the most optimistic material changes in all countries.

The pessimistic projections show an even starker difference between material and non-material futures. Whereas the

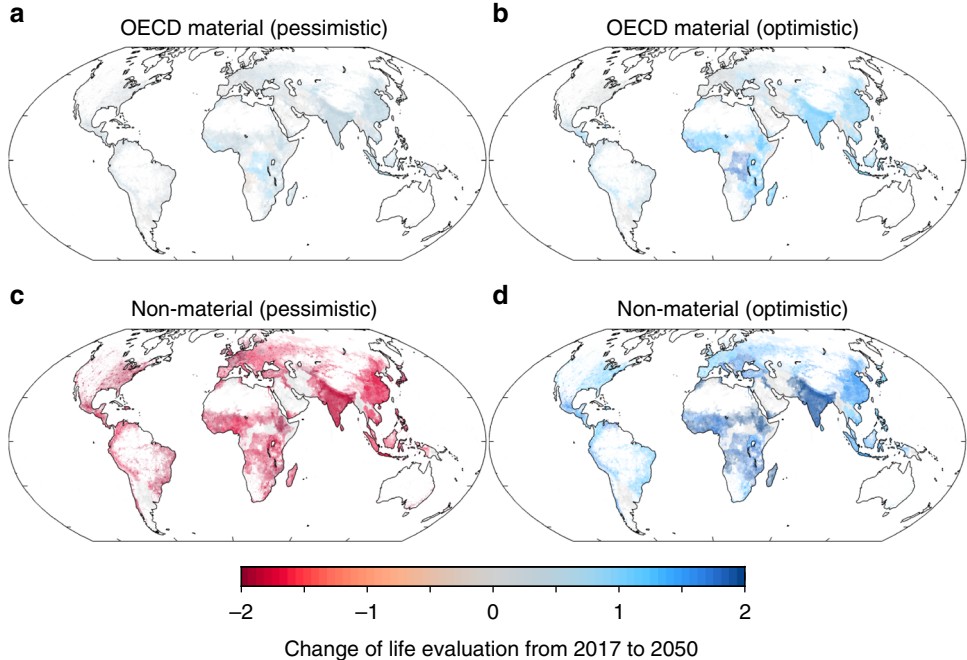

**Fig. 3** Geographic distribution of feasible life evaluation changes. Projections for 2050 are for the OECD material growth scenarios (**a**, **b**), and scenarios in which the non-material predictor variables change at the 10th and 90th percentile rates of recent observations among all countries (**c**, **d**). Coefficients used in the projection are based on the two-period model of life evaluations. The shading on the map is weighted by projected population density

pessimistic OECD projections do not produce negative outcomes in any countries, feasible changes in non-material variables could produce a large decrease in life evaluation in any country by 2050. This reflects the fact that, in recent years, the social variables have had large negative trends in many countries, whereas material variables are generally expected to improve, or at least remain stable everywhere.

## Discussion

Our analysis suggests that large future changes in global life evaluations are feasible, and could be either positive or negative, based on the observed changes of proxy variables within countries during 2005–2016. Our scenarios treat material and non-material proxy variables independently, so that the modeled changes discussed above are additive. For instance, using the OECD and 10/ 90 percentile Non-material Trends, the combined feasible changes give a range of projected 2050 global average life evaluations from 3.9 to 7.6. Regardless of whether the material trends or OECD projections are used, the major part of the summed ranges is due to the non-material factors, because the range of material rates of change is smaller, relative to their modeled impact on life evaluations, than the range of rates of change of the non-material variables. Thus, non-material variables can feasibly make larger impacts than income on a multi-decadal timeframe.

Furthermore, we earlier emphasized that variance-standardized effect sizes are larger for non-material factors than material factors in the 2P dynamic model, while income is the strongest predictor in the cross-sectional model. However, the *raw* coefficients have the opposite relationship, in that the ratio of raw effect size of income to raw effect size of social support (or to any of our other non-material variables) is larger for the 2P dynamic model than the cross-sectional model. Therefore, carrying out projections for long-run changes using coefficients from the cross-sectional model results in even starker dominance by non-material factors.

Speculatively, our model may overestimate the role of GDP per capita for another reason. A well-studied phenomenon first noticed by Easterlin[18] is related to the smaller predictive role of income in explaining long-run variation as compared with annual changes[19]. Human life evaluations appear to adapt to the ambient material affluence over time, in the sense that our mental benefit gradually accommodates to the experienced level[20,21]. This contrasts with social aspects of life, for which there is little evidence of such adaptation[22]. In a second phenomenon, humans respond to material consumption norms set by those around them. Both phenomena help to explain the lack of long-run response of life evaluations at the national level to rising incomes during economic growth, for example, refs. [19,23], and may explain the difference in raw coefficient sizes mentioned above. Furthermore, life evaluations may be sensitive to national (as opposed to local) rank of affluence. Such positional concerns would imply an even smaller long-run direct benefit from aggregate economic growth across all countries, because even the cross-sectional distribution of life evaluations would reflect in part the outcome of a long-run zero-sum game.

In further support of the importance of non-material factors, we point out that, during the period of observations, the global mean of life evaluations did not change significantly, despite an average increase of incomes by 17% and lengthening of life expectancy by 2.6 years (Supplementary Tables 1–3). The global improvements in material conditions, which were accompanied by improvements in average freedom and corruption, were counteracted by a decrease in average social support, and the reported change in global lifeevaluations was negligible.

Although we have focused here on subjective life evaluation, we undertook an identical model-building and projection procedure for affect measures, which capture day-to-day feelings. A predictive model based on the six proxy variables does not capture as much of the variation in affect as the models for life evaluation[8] (see Supplementary Tables 4–7). Nonetheless, with this caveat in mind, the trend-based projections suggest that the average affect balance of humans in 2050 could decrease by as much as 0.28 or increase by as much as 0.31 on a two-point scale within the 90th to 10th percentile trends, spanning 30% of the total range, very

similar to the relative magnitude of feasible changes in life evaluations. Furthermore, the changes in affect are dependent exclusively on non-material factors, with no discernible influence of material changes (see Supplementary Note 2). Thus, large changes in affect are also feasible, could also be positive or negative, and appear to be exclusively dependent on non-material factors.

We would emphasize that the predictor variables we use are proxies, rather than direct mechanistic drivers of life evaluation. There are many dimensions of material supports, such as nutrition, shelter, and infrastructure, which may not vary directly with per capita GDP. Provision of these material needs may also help to build trust, support systems among family and communities, and freedom of choice. Likewise, our non-material proxies do not capture all of the dimensions of social experience known to be important for life evaluations, and a breakdown of these non-material conditions likely imperils the material ones[24]. In order to address the possibility that these interactions are statistically important, we repeated our analysis using models augmented with a series of interaction (also know as moderation) terms between pairs of predictors with statistically significant effects in our 2P model. These tests, which are detailed in Supplementary Note 3 and Supplementary Table 8, suggest a statistically significant interaction between GDP and freedom of choice, but this interaction does not alter our general conclusions nor the greater importance of non-material factors (see Supplementary Figure 1). Nevertheless, we recognize that our results remain limited by the use of a global model with six predictors that are, unavoidably, inter-related. In the future, the accumulation of larger databases could allow the development of a formal model selection process to choose among broader possibilities of predictor variables, and may ultimately allow the development of mechanistic models with greater predictive power.

Despite the simplifications inherent in this first attempt to project future human well-being, our results show that the greatest benefits to be potentially made over the next decades, as well as the most dangerous pitfalls to be avoided, lie in the domain of social fabric. Focusing on income among the effects of long-run policy is therefore too narrow, and misses the majority of the human well-being effects that could feasibly occur, based on past experience. Given that the policies known to support a strong social fabric can differ from those focused on economic growth, our results suggest that scarce resources may be better prioritized towards explicitly social aims if human well-being is the goal.

## Methods

**Survey data**. Following recent World Happiness Reports, we include a measure of income along with variables representing healthy life expectancy (life expectancy), perceived level of corruption (corruption), freedom to choose (freedom), prevalence of donating to others (giving), and availability of informal social support (social support). The English wording of the life evaluation question is "Please imagine a ladder, with steps numbered from 0 at the bottom to 10 at the top. The top of the ladder represents the best possible life for you and the bottom of the ladder represents the worst possible life for you. On which step of the ladder would you say you personally feel you stand at this time?" The social support variable is the national average of dichotomous responses to "If you were in trouble, do you have relatives or friends you can count on to help you whenever you need them, or not?" The freedom variable is the national average of dichotomous responses to "Are you satisfied or dissatisfied with your freedom to choose what you do with your life?" The corruption variable is the national average of dichotomous answers to two questions: "Is corruption widespread throughout the government or not?" and "Is corruption widespread within businesses or not?" In order to isolate the prevalence for giving from the variation in financial capacities, national mean responses to "Have you donated money to a charity in the past month?" are regressed on GDP per capita, and the residual becomes the Giving variable[25]. Further descriptions and important notes for these variables are given in the statistical appendix for Chapter 2 of the 2017 report[8]. Our measure of income is the natural logarithm of internationally comparable (i.e., purchasing power parity) GDP per person; this measure has a relatively good linear fit with life evaluations,

for example, ref. [26]. By considering these measures at the aggregate (country) level, we best capture the sum of individual effects and those which come from public goods (at national and subsidiary scales) and from social and contextual effects at all spatial scales up to the country level. Our use of country-level data is motivated by the assumption that externalities (e.g., positional effects, or public goods) are likely to be large within countries, but are likely to be relatively small between countries.

Summary information about the dataset is provided in Supplementary Note 1, including descriptive statistics of levels (Supplementary Table 3), annual changes (Supplementary Table 2), and 2P changes (Supplementary Table 1) of our key variables. We also provide estimates of pairwise correlations among levels (Supplementary Table 9) and 2P changes (Supplementary Table 10) in our key variables.

**Regression models for well-being (overview)**. We estimate four different models to characterize the relationship between country-level SWB and our six predictor variables. Each model is suitable for estimation using a weighted least-squares approach. One model predicts the relationship among countries, another predicts within-country changes between our two time periods, and two approaches predict year-to-year changes within countries. We use the 2P model in our primary analysis in order to remain conservative with our primary conclusions, and in order to avoid the short-run effects of the global financial crisis (GFC) in our estimates of long-run trends, but we show results from the other models in Supplementary Notes 1 and 3. For completeness, the four models and the standardization are explained below. For each of these models, the corresponding estimates are tabulated in Supplementary Note 2 both as raw coefficients and secondly as unitless standardized coefficients.

As emphasized in the main text, we cannot identify and isolate independent causal relationships between supports (predictor variables) and life evaluations. Rather, our set of proxy variables, with their shared variance, captures one projection of the true underlying processes. In the case of our Giving variable, we err on the side of under-estimating the importance of giving by using the residual from a regression of the underlying donations variable on an income measure. This may in turn result in over-estimating the importance of income.

**Regression models for well-being (static model)**. First, we consider the cross-sectional (XS) relationship

$$S_{it} = a + \sum_{j=1}^{6} b_j x_{jit} + c_t + \nu_i + \varepsilon_{it}, \qquad (1)$$

where $S_{it}$ is the SWB in country $i$ in year $t$; $a$ is a global constant; $x_{jit}$ are the six predictor variables; $c_t$ is a constant in a given year $t$ for all countries; and $\nu_i + \varepsilon_{it}$ is an error term clustered at the country level. Here, $b_j$ are the regression coefficients describing the effects of interest. In estimating (1) we use country weighting because we consider the political–cultural–economic dynamics to be distinct across countries and therefore consider each country to be one sample unit. We cluster errors at the country level to take into account the fact that we have multiple observations (i.e., by year) for each country, but do not capture all country features in the model. Our inclusion of year indicators (dummy variables) $c_t$ is to remove variation related to global secular trends (e.g., average global economic growth) and short-run global cycles (e.g., the GFC), so that our estimated $b_j$ capture the cross-sectional variation.

In order to quantitatively compare these relationships $b_j$ across predictors $j$, we calculate the standardized regression coefficients $\beta_j \equiv \frac{\sigma_j}{\sigma_s} b_j$; that is, we estimate the standardized variable equation

$$\frac{S_{it}}{\sigma_s} = \tilde{a} + \sum_{j=1}^{6} \beta_j \frac{x_{jit}}{\sigma_j} + \tilde{c}_t + \tilde{\nu}_i + \tilde{\varepsilon}_{it}, \qquad (2)$$

where $\sigma_s$ is the standard deviation of $S_{it}$ across all country-year observations, but again calculated with country weights to take into account the fact that not all countries have the same number of observations (years). The standard deviation $\sigma_j$ of $x_{jit}$ is calculated the same way. Thus, $\beta_j$ should be interpreted as the number of standard deviations of change in $S$ associated with one standard deviation change in $x_j$.

**Regression models for well-being (dynamic models)**. While (1) models the variation across countries, we also generate models of changes over time in order to project future changes based on past rates of change. We do this in three ways. Two of them, fixed effects (FE) and first differences (FD), use the year-to-year changes in the predictor variables for each country to explain year-to-year changes in SWB. The third method (2P) considers only the longer-run difference for each country from early in the Gallup World Poll data (2005–2007) to the most recent years (2014–2016). Taking the means during each of these two periods excludes the acute effects of the GFC and its recovery.

**Regression models for well-being (annual changes)**. First, we consider the year-by-year changes, captured in an FE model by including an indicator (constant offset) $d_i$ for each country (this is equivalent to subtracting the mean value for each country):

$$S_{it} = a + d_i + \sum_{j=1}^{6} b_j^{\text{FE}} x_{jit} + \nu_i + \varepsilon_{it}. \qquad (3)$$

Alternatively, we may model the 1-year changes in an FD equation, as follows:

$$\Delta S_{it} = \sum_{j=1}^{6} b_j^{\text{FD}} \Delta x_{jit} + \nu_i + \varepsilon_{it}. \qquad (4)$$

With finite samples, (3) and (4) do not give identical estimates. Moreover, because our panel is not perfectly balanced, some observations are dropped when estimating (4). Because of this, and the inherent higher efficiency of the FE estimator, we favor (3) but show estimates of both.

As described in the Methods section, we wish to be able to compare the estimated effects of one predictor variable to another—that is, to ascribe relative importance to changes in different predictor variables. For the FD estimator, this is straightforward, as we can construct normalized versions of observed 1-year changes, in analogy to (3):

$$\frac{\Delta S_{it}}{\sigma_{\Delta S}} = \sum_{j=1}^{6} \beta_j^{\text{FD}} \frac{\Delta x_{jit}}{\sigma_{\Delta x_j}} + \tilde{\nu}_i + \tilde{\varepsilon}_{it}. \qquad (5)$$

The term $\beta_j$ in (5) explains the distribution of 1-year changes in $S$, such that $\beta_j$ is the number of standard deviations of change in $\Delta S$ associated with 1 s.d. change in $\Delta x_j$, where these variances are calculated across observed 1-year changes.

In order to express the relative importance of estimates from the FE estimator, (3), we use the same standard deviations of FD to transform the $b_j^{\text{FE}}$ in (3) to

$$\beta_j^{\text{FE}} \equiv \frac{\sigma_{\Delta x_j}}{\sigma_{\Delta S}} b_j^{\text{FE}}. \qquad (6)$$

Estimating $\beta_j^{\text{FE}}$ has the advantages over $\beta_j^{\text{FD}}$ mentioned above (efficiency, more inclusion) for FE.

**Regression models for well-being (2P changes)**. The third method constructs an equation with only two observations per country, the early period and the later period (2005–2007 and 2014–2016), in order to model longer-run changes $\delta x_j$ and $\delta S$. In this context, the FE and FD estimators are identical; thus, we write simply

$$\delta S_{it} = \sum_{j=1}^{6} b_j^{2\text{P}} \delta x_{jit} + \nu_i + \varepsilon_{it}. \qquad (7)$$

Analogously to the methods above, we define the standardized coefficients

$$\beta_j^{2\text{P}} \equiv \frac{\sigma_{\delta x_j}}{\sigma_{\delta S}} b_j^{2\text{P}} \qquad (8)$$

using the standard deviation of longer-run changes across countries. This estimate has fewer observations than those modeling annual changes, and the confidence intervals are slightly looser.

As mentioned in the Discussion, we also consider moderation effects between variables in the 2P model by testing pairwise interaction terms, detailed in Supplementary Note 3. We find a significant moderation effect only between GDP and freedom of choice.

**Regression models for well-being (results)**. Estimates of the models for life evaluations are described in Supplementary Note 2 and summarized in Supplementary Table 4. In addition, we report analogous estimates for our other measures of well-being, namely positive affect (Supplementary Table 5), negative affect (Supplementary Table 6), and affect balance (Supplementary Table 7). We also carry out and explain an estimate at the regional level (reported in Supplementary Table 11).

**Recent trends (overview)**. We define future scenarios by projecting different quantiles of recent country trends in predictor variables into the future. Rather than defining complex and arbitrary story lines, we represent the range of feasible outcomes by defining idealized scenarios. We denote the total feasible range of possible rates of change as that encompassed by the 10th to 90th percentile observed range among all countries. Thus, we use the assumption that, if a given

rate of change in a proxy variable was observed in 10% of all countries over the 11-year observation period, it is feasible that the variable could change at this rate in any country over the next three decades. This differs from the full range of possible changes, for which we have no first principles understanding, but which is likely to be greater than our feasible range given the possibility of unforeseeable, radical transformation. To better characterize the distribution of likely futures, we also make projections using the 30th, 50th, and 70th percentiles of recent trends, but we focus on the 10th and 90th percentiles for our primary analysis.

For the purpose of illustrating large changes in our predictor variables, Supplementary Figure 6 shows recent trends (with 3-year smoothing) in the nine countries which exhibited the most variation in each variable.

The estimation of recent trends in our predictor variables and in SWB is complicated by two issues. First, the panel of available data is incomplete; that is, there are data gaps for some country-year variable entries. Table 1 shows the number of countries with data by year or period for our various approaches. In order to estimate trends that are not biased by a changing composition of countries, a subset of countries with complete data can be used, or data can be appropriately weighted in order to account for features of the incomplete panel. Secondly, the GFC had a significant impact on some countries in 2008, yet these effects were somewhat temporary. The short-term changes following the GFC may not contribute constructively towards estimating the longer-run changes that might inform scenarios to 2050. Fortunately, the Gallup World Poll time series is now long enough that estimates can be made even after excluding several years following the start of the GFC.

**Recent trends (country estimates)**. We therefore consider the following three estimation models of recent time trends in our key variables:

First, a differences model: Take averages for each country for each period (2005–2007, 2014–2016), and calculate the difference. This avoids the GFC, but entails a two-step process to calculate confidence intervals for the differences.

Second, a 2P regression, excluding GFC model: A one-step method which provides estimates of confidence intervals is to carry out a regression

$$x_{it} = x_{i0} + B_i \pi_t + \varepsilon_{it}, \qquad (9)$$

where $\pi_t$ is an indicator variable for periods after/before the GFC:

$$\pi_t = \begin{cases} 1, & t \geq 2014, \\ 0, & \text{otherwise} \end{cases} \qquad (10)$$

and where observations are estimated with weights

$$w_{it} = \begin{cases} \frac{1}{N^i_{t \leq 2007}}, & t \leq 2007, \\ 0, & 2008 \leq t \leq 2013, \\ \frac{1}{N^i_{t \geq 2014}}, & t \geq 2014. \end{cases} \qquad (11)$$

Here $N^i_{t \leq 2007}$ is the number of observations of country $i$ in the early period. The point estimates of $B_i$ in (9) are identical to the difference in period means for country $i$, but the standard error is calculated correctly.

In order to convert the estimates $B_i$ of change between the two periods to an estimate of the annual rate of change, we divide by 9, the number of years between the middles of the two periods.

Third, and lastly, an all-year (annual) regression model: The annual rate of change can be estimated for country $i$ and variable $x$ using all years $t$ for which $x_{it}$

---

**Table 1 Completeness of panel data**

| Year | Any data | All data | Balanced ≥2007 | Any <2008 | Any >2013 | Any (2P) |
|------|----------|----------|----------------|-----------|-----------|----------|
| 2005 | 27 | 21 | | 118 | | 109 |
| 2006 | 85 | 77 | | 118 | | 109 |
| 2007 | 99 | 92 | 64 | 118 | | 109 |
| 2008 | 104 | 100 | 64 | | | |
| 2009 | 109 | 105 | 64 | | | |
| 2010 | 118 | 110 | 64 | | | |
| 2011 | 138 | 130 | 64 | | | |
| 2012 | 134 | 122 | 64 | | | |
| 2013 | 130 | 119 | 64 | | | |
| 2014 | 137 | 122 | 64 | | 129 | 109 |
| 2015 | 136 | 120 | 64 | | 129 | 109 |
| 2016 | 133 | 116 | 64 | | 129 | 109 |

The first column lists the number of countries with coverage within our core set of variables for each year of the Gallup World Poll, after some imputation done in accordance with the data appendix of Helliwell et al.[8]. The All data column lists the number of countries with all of our core variables. The Balanced column shows the number of countries in the largest balanced (complete, rectangular) panel covering all years after 2006. For our two-period model, we use the countries with at least one observation in the early period and at least one observation in the later period

**Table 2 Estimated recent global trends and distribution**

|  | (1) Life today | (2) −Affect | (3) +Affect | (4) Log(GDP/capita) | (5) Life expectancy | (6) Giving | (7) Freedom of choice | (8) Corruption | (9) Social support |
|---|---|---|---|---|---|---|---|---|---|
| Global trend (country weights): 2P | .003 | **.002**† | .0002 | **.018**† | **.29**† | −.0004 | **.004**† | −.002 | −.002* |
|  | (.005) | (.0006) | (.0005) | (.002) | (.020) | (.001) | (.0010) | (.0009) | (.0006) |
| Global trend (country weights): annual | −.002 | **.002**† | −8e−05 | **.015**† | **.27**† | −.0006 | **.003*** | −.002 | −.002† |
|  | (.006) | (.0006) | (.0006) | (.002) | (.022) | (.001) | (.001) | (.0009) | (.0007) |
| Global trend (pop'n weights): 2P | −.006 | .002 | .0002 | **.037**† | **.25**† | −.0006 | **.004*** | −.003* | −.0007 |
|  | (.023) | (.002) | (.0008) | (.009) | (.023) | (.002) | (.001) | (.001) | (.0009) |
| Global trend (pop'n weights): annual | −.012 | .003 | −.0003 | **.033*** | **.22**† | −.002 | **.003** | −.004 | −.001 |
|  | (.025) | (.002) | (.0009) | (.011) | (.026) | (.003) | (.002) | (.001) | (.0009) |
| Fraction p ≤.05: 2P | .32 | .23 | .18 | .76 | 1.00 | .27 | .27 | .34 | .21 |
| Fraction p ≤.05: annual | .29 | .31 | .25 | .81 | .98 | .30 | .33 | .31 | .26 |
| 10th %ile: 2P | −.073 | −.005 | −.008 | −.003 | .12 | −.016 | −.008 | −.013 | −.008 |
| 10th %ile: annual | −.099 | −.005 | −.009 | −.004 | .12 | −.018 | −.010 | −.018 | −.012 |
| 30th %ile: 2P | −.017 | −.002 | −.002 | .008 | .16 | −.006 | −.002 | −.006 | −.003 |
| 30th %ile: annual | −.037 | −.0006 | −.003 | .009 | .16 | −.007 | −.002 | −.009 | −.004 |
| 50th %ile: 2P | .006 | .001 | .0006 | .017 | .22 | −.001 | .002 | −.003 | −.001 |
| 50th %ile: annual | −.0002 | .003 | .0009 | .020 | .23 | −.003 | .004 | −.003 | −.001 |
| 70th %ile: 2P | .034 | .005 | .003 | .028 | .32 | .004 | .011 | .001 | .0006 |
| 70th %ile: annual | .031 | .009 | .005 | .029 | .33 | .003 | .011 | .002 | .003 |
| 90th %ile: 2P | .077 | .010 | .009 | .043 | .55 | .016 | .017 | .010 | .006 |
| 90th %ile: annual | .086 | .018 | .012 | .047 | .51 | .017 | .021 | .011 | .010 |

The first four rows show estimated global trends for the period 2005–2016 and between the periods 2005–2007 and 2014–2016. The 2P versions of the five quantiles of the across-countries distributions are used to build our future scenarios
Significance: †**.1%**, ***1% 5%**, 10%

was observed, according to the following equation:

$$x_{it} = a_i + b_i t + \varepsilon_{it}. \tag{12}$$

**Recent trends (global and regional estimates).** For directly estimating global (or regional, multi-country) trends, we follow a similar procedure as the 2P and annual approaches described above, but using two alternative approaches to weighting observations. In one, we use weights which count each country equally, and in another we multiply the weights described previously by country populations. Thus, we have four variants of estimated global trends: for each variable there is a 2P version and an annual version, and for each of those there are country weights and population weights.

By contrast, we consider countries as individual observations when thinking about the global distribution of trends, and thus calculate, for example, the 10th percentile trend from the unweighted set of countries.

**Recent trends (results).** We estimate the 2P and annual models for each country, and for the world as a whole. Table 2 present the results. In general, we find excellent agreement between the 2P and annual estimates. Although the annual estimates use more observations and should therefore be more efficient, they have slightly higher standard errors, likely reflecting the short-term variation during the GFC.

Overall, the trend in life evaluations, measured as per-year change on an 11-point scale, is very small (insignificant). The material variables income and life expectancy are globally increasing significantly, according to all our specifications. There are also significant global trends in the non-material variables, whose changes are measured as annual change on a 0–1 scale. Freedom and corruption are both improving in all specifications, while social support is getting worse (diminishing), and significantly so averaged across countries. Thus, the predictor variables are significantly trending but not all in an optimistic or pessimistic direction.

Table 2 also reports, for each variable, the fraction of countries with statistically significant ($p \leq .05$) rates of change. These fractions are in all cases considerably >5%, suggesting that trends, both positive and negative, are sustained (and real). This fact is central to our argument in that it supports our assumption that we can gauge reasonable rates of change based on the distribution of changes in the recent past. The remaining rows of Table 2 show the 10th and 90th, and intermediate, percentiles from these distributions. We consider these to correspond to plausible optimistic and pessimistic trends in the future scenarios. In general, these values are more extreme for the annual model than the 2P model, and we use the latter in order to remain conservative in our scenarios.

**Future scenarios.** We present three sets of scenarios, each with more and less optimistic and pessimistic cases. Two sets focus on changes in material circumstances—the OECD macroeconomic projections, and our material variable trends, which project 10th, 30th, 50th, 70th, and 90th percentile rates of change for all countries. The third scenario pair projects the same quantiles of rates of change for

our non-material predictor variables. Note that future changes in predictor variables could potentially occur more rapidly than has occurred in any country over the observed timeframe, but our approach focuses on the most likely range based on past experience.

The OECD projections use two global economic scenarios, devised to explore possible futures for major environmental challenges, including climate change. From these Shared Socioeconomic Pathways[27], we bracket the range of possibilities for income and population growth by selecting the two most contrasting scenarios, the rapid population growth, slow economic growth, weak convergence (SSP3) and the slow population growth, rapid economic growth, strong convergence (SSP5) as our OECD pessimistic and OECD optimistic Material trends scenarios, respectively.

We project each of the sets of trends forward to 2050, starting from 2016 values in each country. Country level means derived from dichotomous variables have natural saturation points at 0 and 1; otherwise, there are no restrictions on projected values. We use OECD projections of population change by country in order to calculate globally averaged life evaluations for all humans.

To test the robustness of our findings, we undertake each of our projections using alternative models. Tabular summary data for the three sets of scenarios using 2P, FE, and XS coefficients are given in Supplementary Tables 12 and 13, and Supplementary Table 14, respectively. These same values are also presented in summary plots in Supplementary Figure 7, for both weighted and unweighted means.

We also provide maps to represent the population-weighted projections in our scenarios using our three different sets of coefficients. Supplementary Figure 2 shows the same cases as in our main text, which use the 2P coefficients, but includes in the middle row the material trend scenarios. They are not included in the main text due to their geographic uniformity. Supplementary Figure 3 and 4 show our projections under the alternative specifications of XS and FE coefficients, respectively. Finally, Supplementary Figure 5 shows projections for affect balance, using the 2P coefficients.

**Code availability.** No special code was used in our analysis, which proceeded as described in our methods. Requests for scripts needed to reproduce our results will be reviewed and made available on a case-by-case basis by the corresponding author.

## Data availability

Our country-mean annual time series data come from the 2017 World Happiness Report. Its Statistical Appendix[8] contains annual country averages of several variables for the entire span of the Gallup World Poll, which began with partial coverage in 2005. Helliwell et al.[8] provide further information on the data and on possible imputation methods to fill gaps in some of the coverage.

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

## Acknowledgements

C.B.-L. acknowledges financial support from Canada's Social Sciences and Humanities Research Council grant 435-2016-0531. E.D.G. acknowledges financial support from the European Research Council (ERC) under the European Union's Horizon 2020 research and innovation programme (grant agreement no. 682602), and from the Spanish Ministry of Science, Innovation and Universities, through the María de Maeztu program for Units of Excellence (MDM-2015-0552).

## Author contributions

C.B.-L. and E.D.G. designed the research. C.B.-L. performed the data analysis and modeling. C.B.-L. and E.D.G. wrote the paper.

## Additional information

**Competing interests:** The authors declare no competing interests.

