## [Peer Review File · Nature Communications]

Reviewers' comments:

Reviewer #1 (Remarks to the Author):

Review of Feasible future Scenarios for Future Life Evaluations

I like this paper a lot because of its focus on where we could be in terms of well-being in the future, 2050, and what is

most important in producing good or bad trends. Novel and interesting.

A couple of suggestions:

1. The authors use Belgium and Egypt to anchor our judgments. Not so meaningful to many or most readers. Belgium? who knows? Egypt-- well they are in a bit of turmoil, but where do they stand compared to the worst places. So the well-being anchors don't anchor me much.

2. It would be worth noting that enjoying life, Positive affect, might be more influenced by those social factors than life satisfaction. Those measures might show even bigger differences.

3. The authors use the 10th and 90th percentiles to show how bad or good things possibly could be. But those both seem

unrealistic in that most nations are probably not going to rise or fall to those levels. I would also like to see some more

realistic alternatives such as 60-40 or 70-30, where we can imagine possible scenarios where these would likely occur.

Reviewer #2 (Remarks to the Author):

Using the Gallup World Poll data, this paper investigates the correlation between six variables (four for non-material life and two for material life) and subjective well-being, and then predicts subjective well-being to the year 2050 based on the observed historical trends in the six variables. The main results are that projected effects of material factors only lead to modest changes in global life evaluations, while in contrast, projected non-material effects lead to large changes. The projection is novel. The statistical analysis which largely follows the World Happiness Report is convincing. The highlights on the importance of non-material effects are of great interest to researchers and policy makers. In summary, the paper is well written.

One thing I'd like to see is the projected life evaluation by 2050 with neutral trends, e.g. using the middle value (50th percentile) of changes in material and non-material variables. It would complement the current analysis with optimistic and pessimistic trends.

Reviewer #3 (Remarks to the Author):

This is a highly interesting manuscript, dealing with important variables and relationships. That data show that cross-sectional and longitudinal changes in life satisfaction at the national level are associated strongly with material and non-material factors (replicating previous work), and that predicted changes in these factors impact on projected country-level well-being much more for non-material than material factors, largely due to the greater variability in non-material factors. This latter argument and analysis is novel and of great interest to a range of readers.

The complex statistical modelling as presented here is not entirely familiar to me, but in multi-level models I do normally expect to see not only group-level coefficients (here, nation-level) but also coefficients for the individual-level relationships, and the cross-level effects. Is this information omitted to simplify the models, given the additional time variables that already introduce complex additional comparison models to the paper? This is perhaps understandable, but reporting the population-weighted coefficients is no substitute, in my mind, to reporting a full multi-level model.

I also would like to see acknowledgement that interpretation of the unique relationships of IVs in these regressions does not take into account the shared variance among them, which is substantial. Among the IVs, only for giving is this inter-relationship dealt with statistically (by using for the construct 'giving' the residual or disproportionate giving after controlling for income); this choice gives the full shared variance to the income variable. This should be acknowledged – first, it may be that the positive role for income is exaggerated due to this approach, while the role of giving is under-estimated by measuring it as disproportionate giving above the level expected by income. But more broadly, the mediation and suppression patterns in this data would be over-looked, while only direct associations are considered. The discussion must acknowledge this complexity, even if the

analyses do not expand to consider theoretically interesting indirect effects (e.g., longevity to giving or income to corruption).

I also consider that in the bigger picture, the interactions (moderation) between economic and non-material factors are surely of great interest, rather than only the present comparison of the impact of one or the other class of variables, as if they are not highly inter-related. For example, it may be that the impact of non-material factors is greater once a threshold of economic growth or of GDP has been achieved, or is blunted (or heightened) when economic recessions occur. The period here of economic tumult in the GFC would allow you to explore this type of interaction, and I hope that the authors will consider this, if not in the present paper, then in a future one.

With those caveats, it is certainly the case that the core message of this paper – that changes in non-material factors have strong unique consequences for well-being that policy-makers do not routinely consider and yet should target – will be of interest to readers from a range of disciplines, and should be published.

Missing information:

The SI should give more details of the method especially for the key DV – at present the text only reads “Our main text focuses on one primary measure of subjective well-being (SWB), namely the cognitive evaluation of life question called “Cantril’s Ladder.” (p. 14)

Within the SI, for key variables, the text reads “Detailed descriptions for these variables are given in the si.” (!?) p. 13 – but these are missing.

Minor points:

Figure 1 : Widen y axis to show full confidence interval range.

Suggest replacing ‘xs’ and ‘2p’ with more intelligible labels, such as cross-sectional and pre/post GFC.

Winnifred Louis

Reviewers' comments:

Reviewer #1 (Remarks to the Author):

Review of Feasible future Scenarios for Future Life Evaluations

I like this paper a lot because of its focus on where we could be in terms of well-being in the future, 2050, and what is most important in producing good or bad trends. Novel and interesting.

A couple of suggestions:

1. The authors use Belgium and Egypt to anchor our judgments. Not so meaningful to many or most readers. Belgium? who knows? Egypt-- well they are in a bit of turmoil, but where do they stand compared to the worst places. So the well-being anchors don't anchor me much.

authors' response:

Thank you for this feedback. We have addressed this comment in three ways:

1. We offer more than one comparison country, in each case.
2. We report explicitly where Belgium and Costa Rica and Egypt and India ranked globally in the WHR in 2016.
3. We no longer refer to individual countries in the abstract, but instead to percentile ranges.

The new text segments are thus:

... 6.9, which for comparison is the level actually reported by Belgium and Costa Rica in 2016, when they had the 18th and 14th highest life evaluations. Conversely, if unfavorable observed non-material trends were to prevail in all countries, our projections suggest that the drop in life evaluations by 2050 would take the global average to 3.4, below the level of Egypt and India in 2016, when they ranked 118th and 120th out of 157 countries.

in the main text and

In contrast, projected non-material effects lead to future scenarios with global average life evaluations covering more than a three-fold wider range, lying anywhere from the present day top 15% of countries to the present-day bottom 25% of countries.

in the abstract.

2. It would be worth noting that enjoying life, Positive affect, might be more influenced by those social factors than life satisfaction. Those measures might show even bigger differences.

authors' response: Thank you for raising the issue of future changes in affect. In fact, the SI already includes a complete set of analysis of affect measures, parallel to the main text's treatment of life evaluations. Tables S7, S8, and S9 provide model estimates for positive and negative affect and for affect balance. In addition, for the cross-sectional case, this question is discussed in the World Happiness Reports. According to our estimates, it turns out that the social factors are as much protective against negative affect as they are supportive of positive affect, but the effects are different for different variables, and gaining further insight into these differences represents research still to be done. As compared with cognitive life evaluations, affect measures certainly have more unexplained

international variance and possibly cultural influence, in addition to being more subject to short-term influences than the enduring factors we focus on in this paper. Figure S6 shows our scenario maps for the case of affect balance, using 2P coefficients, and our findings hold as strongly for this measure as they do for life evaluations.

We previously mentioned the supplementary analysis and findings in the discussion section of the main text, but recognize that it was not very prominent and was easily

missed. We have therefore revised the text to better highlight the affect analysis and its conclusions, which are very well in agreement with the reviewer's intuition.

3. The authors use the 10th and 90th percentiles to show how bad or good things possibly could be. But those both seem unrealistic in that most nations are probably not going to rise or fall to those levels. I would also like to see some more realistic alternatives such as 60-40 or 70-30, where we can imagine possible scenarios where these would likely occur.

authors' response: Thank you for this very helpful suggestion. We have added indications to Figure 2 of outcomes from projections based on the 30th, 50th, and 70th percentile trends, in addition to the full 10th–90th %ile ranges that were shown previously. Relevant portions of the text were modified in order to be consistent with the additional scenarios. Our conclusions and interpretation all hold under these more moderate projections. If anything, they further highlight the optimistic nature of the OECD projections as compared with the experience of recent material trends.

3 Reviewer #2 (Remarks to the Author):

Using the Gallup World Poll data, this paper investigates the correlation between six variables (four for non-material life and two for material life) and subjective well-being, and then predicts subjective well-being to the year 2050 based on the observed historical trends in the six variables. The main results are that projected effects of material factors only lead to modest changes in global life evaluations, while in contrast, projected non-material effects lead to large changes. The projection is novel. The statistical analysis which largely follows the World Happiness Report is convincing. The highlights on the importance of non-material effects are of great interest to researchers and policy makers. In summary, the paper is well written. One thing I'd like to see is the projected life evaluation by 2050 with neutral trends, e.g. using the middle value (50th percentile) of changes in material and non-material variables. It would complement the current analysis with optimistic and pessimistic trends.

authors' response: Thank you too for this important suggestion. We have addressed it under our responses to Reviewer #1. The median values help to clarify the nature of the broad ranges we show.

4 Reviewer #3 (Remarks to the Author):

This is a highly interesting manuscript, dealing with important variables and relationships. That data show that cross-sectional and longitudinal changes in life satisfaction at the national level are associated strongly with material and non-material factors (repl-cating previous work), and that predicted changes in these factors impact on projected country-level well-being much more for non-material than material factors, largely due to the greater variability in non-material factors. This latter argument and analysis is novel and of great interest to a range of readers.

The complex statistical modelling as presented here is not entirely familiar to me, but in multi-level models I do normally expect to see not only group-level coefficients (here, nation-level) but also coefficients for the individual-level relationships, and the cross-level effects. Is this information omitted to simplify the models, given the additional time

variables that already introduce complex additional comparison models to the paper? This is perhaps understandable, but reporting the population-weighted coefficients is no substitute, in my mind, to reporting a full multi-level model.

authors' response: We agree that for the fullest understanding of the mechanism and scales of life satisfaction supports, multi-level modeling is appropriate. Although one of us has published in the past using respondent-level data from the Gallup World Poll (and see also recent World Happiness Reports for such analyses), we now explain in the first section of the SI why country-level analysis is appropriate for country-level projections. To make aggregate projections, we need to add to the individual effects the sum over all of the positive and negative contextual effects over neighbourhood, community, city, and regional levels in each country, and over the effects of public good provision at each government level within each country. There may be contextual effects / spillovers across national borders as well, but these must largely be neglected (and have been so far in the literature) given one planet Earth. This summing up is accomplished, and appropriately weighted, up to the country level simply by modeling country-level changes in life evaluations as being driven by country-level changes in predictor variables.

As an aside, there is a second advantage to the macro-level analysis. The cost for access to the individual-level data from Gallup would be USD\$30,000, which is prohibitive for most interested parties. Our method appropriately uses the macro data published in the World Happiness Report and is therefore reproducible using solely open-data.

I also would like to see acknowledgment that interpretation of the unique relationships of IVs in these regressions does not take into account the shared variance among them, which is substantial. Among the IVs, only for giving is this inter-relationship dealt with statistically (by using for the construct 'giving' the residual or disproportionate giving after controlling for income); this choice gives the full shared variance to the income variable. This should be acknowledged --- first, it may be that the positive role for income is exaggerated due to this approach, while the role of giving is under-estimated by measuring it as disproportionate giving above the level expected by income.

authors' response: Thank you for pointing this out. We have now added another paragraph in the SI, which reads as follows:

In all of our models, as emphasized in the main text, we cannot identify and isolate independent causal relationships between supports (predictor variables) and life evaluations. Rather, our set of proxy variables, with their shared variance, capture one projection of the real interrelated effects. In the case of our "giving" variable, we err on the side of under-estimating the importance of giving by using the residual from a regression of the underlying donations variable on an income measure. This may in turn result in over-estimating the importance of income. In light of our main conclusions, these act as conservative biases.

This emphasizes a point which we feel we have also acknowledged, more generally, in the second last paragraph of the main text.

But more broadly, the mediation and suppression patterns in this data would be overlooked, while only direct associations are considered. The discussion must acknowledge this complexity, even if the analyses do not expand to consider theoretically interesting indirect effects (e.g., longevity to giving or income to corruption).

authors' response: Thank you. We agree that mention of these complexities is an

important inclusion. We discuss these possible relationships, and possible two-directional confounding between material and non-material domains, in the second last paragraph of the main text.

I also consider that in the bigger picture, the interactions (moderation) between economic and non-material factors are surely of great interest, rather than only the present comparison of the impact of one or the other class of variables, as if they are not highly inter-related. For example, it may be that the impact of non-material factors is greater once a threshold of economic growth or of GDP has been achieved, or is blunted (or heightened) when economic recessions occur. The period here of economic tumult in the GFC would allow you to explore this type of interaction, and I hope that the authors will consider this, if not in the present paper, then in a future one.

authors' response: Thank you for this important suggestion. We agree both that this is worthwhile and that it would be an excellent topic for future work.

With those caveats, it is certainly the case that the core message of this paper --- that changes in non-material factors have strong unique consequences for well-being that policy-makers do not routinely consider and yet should target --- will be of interest to readers from a range of disciplines, and should be published.

Missing information: The SI should give more details of the method especially for the key DV --- at present the text only reads "Our main text focuses on one primary measure of subjective well-being (SWB), namely the cognitive evaluation of life question called "Cantril's Ladder." (p. 14)

Within the SI, for key variables, the text reads "Detailed descriptions for these variables are given in the si." (!?) p. 13 --- but these are missing.

authors' response: Thank you for pointing out this shortfall of details and the circular reference, which we have corrected. We have greatly expanded the SI, including explicit mention of where the detailed definitions of the World Happiness Report variables can be found, and provide the exact survey questions behind the life evaluation, social support, freedom, corruption, and giving variables. This new paragraph is the second paragraph in Section S1 of the S.I.

Minor points: Figure 1 : Widen y axis to show full confidence interval range. Suggest replacing 'xs' and '2p' with more intelligible labels, such as cross-sectional and pre/post GFC.

authors' response: Thank you. We have accepted these suggestions; the plots in Figure 1 are revised accordingly.

Reviewers' comments:

Reviewer #2 (Remarks to the Author):

(This reviewer only left remarks to the editor)

Reviewer #3 (Remarks to the Author):

I was reviewer #3 of the original manuscript.

1 I raised a concern that the data is not analysed using MLM. I accept the authors' point about the logistical (cost) benefit of using only the open-access country-level, data. However, I would have liked to see the limitation of the use of country-level-only analysis acknowledged and engaged within the paper.

2 The question of shared variance is addressed only with regard to one pair of IVs, in the SI, and not acknowledged within the main paper. The authors refer to the second to last para of the main text, but I see no acknowledgement of this point there? Particularly with regard to the role of income, I think the interpretation of the present results would have benefitted from this point explicitly being addressed. And the issue applies not just to this pair of IVs, but to the full range of the variables, which show significant inter-relationships as one would expect.

3 And the moderation analysis has not been carried out, examining the interaction of material and social forces, nor is this issue addressed.

I defer to the Editor regarding whether a contribution has been made of sufficient magnitude for this journal. I think the key message of the paper is of potential interest to a wide range of readers. However, the analysis and interpretation gloss over important concerns and limitations, and for this reason I would likely not accept the paper in its present form.

Reviewer #4 (Remarks to the Author):

This paper presents projections of subjective well-being (SWB) throughout the world in 2050 using Gallup Global data from the World Happiness Report and offers insights regarding how both

macroeconomic and non-material or psychosocial factors contribute to these trajectories of SWB into the future. I think this piece takes a novel and interesting approach to the study of SWB, offering projections into the future, which is a unique endeavor in this field, and has important implications for understanding and shaping societal SWB moving forward. Beyond being of great interest to well-being scholars, the finding that psychosocial factors alter projections of future SWB to a greater extent than material factors is an important finding to disseminate to policy makers whose focus is typically restricted to material factors. The authors thoroughly addressed the feedback points of the previous reviewers and I believe this piece offers a strong contribution to the study of SWB.

One very small revision suggestion for a quick clarification is to make the labels for each of your scenario models more consistent throughout the paper. Namely, I found the use of the terms “material world” and “non-material world” in Figures 2 & 3 and the text regarding these models somewhat confusing as “world” can be interpreted to suggest that you were looking at different countries in each of these analyses (especially alongside the map figures) whereas you were actually looking at different predictor variable scenarios. On page 3 the language introducing these models is inconsistent, shifting from “material trends” to “material world” to “non-material scenario” to “non-material trends”—I think it would be helpful to use “(non-)material trends” consistently throughout instead of “world.”

1 Reviewer #2 (Remarks to the Author):

(This reviewer only left remarks to the editor)

2 Reviewer #3 (Remarks to the Author):

I was reviewer #3 of the original manuscript.

I raised a concern that the data is not analysed using MLM. I accept the authors' point about the logistical (cost) benefit of using only the open-access country-level, data. However, I would have liked to see the limitation of the use of country-level-only analysis acknowledged and engaged within the paper.

AUTHORS' RESPONSE: We regret that, in the prior exchange, we gave the impression that the use of country-level-only data is an important limitation in our work. In fact, the cost of the proprietary individual-level data is not the primary reason for using country-level analysis since, without a complete set of sub-national trends (which the micro-data would not provide due to their sampling frame and sample size), our current approach would remain the most appropriate even with micro data in hand.

To better explain why the country-level data is well-suited to our aims, consider that previous work has shown very large externalities *within* countries that affect individuals' life satisfaction. Therefore, were we to analyze effects at finer resolution within a country, or to consider both individual and country-level effects, we would then have the problem of re-aggregating the effects up to a level which subsumes the dominant externalities. Two examples of this type of externality are (1) consumption comparison effects between individuals and their nearby peers, and (2) the funding of public goods, which spreads benefits to many from the income of each individual. Statistically speaking, the optimal way to do that aggregation would be to estimate the total effects at the higher level, which is in fact what we have done. In essence, our approach accommodates spillovers from our predictor variables within countries (which are likely to be large) and assumes that spillovers between countries are small. As we point out in the discussion, it is possible that material consumption reference effects may even hold at the international level [Barrington-Leigh, 2012], in which case our main conclusions would again be understated.

We have now further revised the text in order to mention these considerations in the methods section M.1:

By considering these measures at the aggregate (country) level we best capture the sum of individual effects and those which come from public goods (at national and subsidiary scales) and from social and contextual effects at all spatial scales up to the country level. Our use of country level data is motivated by the assumption that externalities (e.g. positional effects, or public goods) are likely to be large within countries, but are likely to be relatively small between countries.

Despite the appropriateness of the country-level data for this first attempt to project future well-being, we agree that the limitations of our simple approach could be more prominently acknowledged in the manuscript. We have therefore added additional text to the discussion to highlight these limitations, as well as to suggest some means by which future work could improve on our projections.

We feel that, ultimately, increasing availability of richer data in support of more mechanistic models may facilitate far better insight on possible future pathways of future well-being. Developing these models may be a long and challenging process, but the potential benefits to humanity could be tremendous.

2 The question of shared variance is addressed only with regard to one pair of IVs, in the SI, and not acknowledged within the main paper. The authors refer to the second to last para of the main text, but I see no acknowledgement of this point there?

AUTHORS' RESPONSE: The second last paragraph of the main text was:

We would emphasize that the predictor variables we use are proxies, rather than direct mechanistic drivers of life evaluation. There are many dimensions of material supports, such as nutrition, shelter, and infrastructure, which may not vary directly with GDP. Provision of these material needs may also help to build trust, support systems among family and friends, and freedom of choice. Likewise, our non-material measures do not capture all of the dimensions of social experience known to be important for life evaluations, and a breakdown of these non-material conditions likely imperils the material ones [Knack and Keefer, 1997].

The third and fourth sentences point out the inter-relationships and potential shared variance among our set of predictor variables, stating that the material variables may all be correlated with non-material benefits, and vice versa. We have now expanded this discussion to include testing of moderation effects (see below), to properly reflect the limitations of our analysis.

Particularly with regard to the role of income, I think the interpretation of the present results would have benefitted from this point explicitly being addressed. And the issue applies not just to this pair of IVs, but to the full range of the variables, which show significant inter-relationships as one would expect.

3 And the moderation analysis has not been carried out, examining the interaction of material and social forces, nor is this issue addressed.

AUTHORS' RESPONSE: We are grateful to the reviewer for raising this again, as we now recognize that we did not address it sufficiently in the prior revision, and agree that it is an important analysis to include.

In the initial review, we did not completely grasp the reviewer's point (in the jargon we are familiar with this would be called a test for interaction effects among the predictor variables). We have now carried out the suggested analysis, as described in a new section in the SI (Supplementary Note 3). We have also extended the second-last paragraph in the Discussion to raise this subject explicitly and refer the reader to the SI.

Interestingly, our formal test for the significance of these interaction effects fails to find an effect except in one case, which suggests an interaction between GDP and freedom of choice. (The interaction term is negative, suggesting countries with lower values of one variable experience stronger benefits from the other being higher.) We repeated our projection analysis using the model with this interaction term, but found no significant difference from our standard 2P model. We also considered a model with multiple interaction terms, to accommodate all material/non-material pairwise interactions among predictor variables with statistically significant effects. Our findings were also qualitatively robust to using this version of the model. Given the relatively small sample size

(119 countries) we continue to focus on the 6-variable 2P model in the main text, rather than using one of the less-constrained models.

3 Reviewer #4 (Remarks to the Author):

This paper presents projections of subjective well-being (SWB) throughout the world in 2050 using Gallup Global data from the World Happiness Report and offers insights regarding how both macroeconomic and non-material or psychosocial factors contribute to these trajectories of SWB into the future. I think this piece takes a novel and interesting approach to the study of SWB, offering projections into the future, which is a unique endeavor in this field, and has important implications for understanding and shaping societal SWB moving forward. Beyond being of great interest to well-being scholars, the finding that psychosocial factors alter projections of future SWB to a greater extent than material factors is an important finding to disseminate to policy makers whose focus is typically restricted to material factors. The authors thoroughly addressed the feedback points of the previous reviewers and I believe this piece offers a strong contribution to the study of SWB.

One very small revision suggestion for a quick clarification is to make the labels for each of your scenario models more consistent throughout the paper. Namely, I found the use of the terms “material world” and “non-material world” in Figures 2 & 3 and the text regarding these models somewhat confusing as “world” can be interpreted to suggest that you were looking at different countries in each of these analyses (especially alongside the map figures) whereas you were actually looking at different predictor variable scenarios. On page 3 the language introducing these models is inconsistent, shifting from “material trends” to “material world” to “non-material scenario” to “non-material trends”—I think it would be helpful to use “(non-)material trends” consistently throughout instead of “world.”

AUTHORS’ RESPONSE: We thank the reviewer for the supportive appraisal and helpful suggestion. We have removed the word “world” and used consistent terminology throughout, as suggested.

References

- Christopher Barrington-Leigh. Quantity or quantile? A global study of income, status, and happiness. *McGill working paper*, 2012. URL <http://wellbeing.ihsp.mcgill.ca/publications/Barrington-Leigh-DRAFT-ordinalIncome.pdf>.
- Stephen Knack and Philip Keefer. Does social capital have an economic payoff? a cross-country investigation. *The Quarterly journal of economics*, 112(4):1251–1288, 1997.

REVIEWERS' COMMENTS:

Reviewer #3 (Remarks to the Author):

I reviewed this paper (and the revision) as R3, and I appreciate the authors' detailed and constructive engagement with the feedback.

With regards to MLM, I do disagree with the conclusion (that omitting MLM is a plus) but I appreciate the nuance provided in the discussion and I think the authors argue their case well (including that these analyses can be pursued in future research).

The extra discussion of shared variance is important, and I very much appreciate the test of interactions (or moderation; sorry for the confusion in my earlier review). I think this section of the supplemental materials itself may stimulate a great deal of interest! Normally I would expect to see significant interactions followed up with tests of the simple slopes (e.g., of freedom of choice for high and low income, or vice versa), but this can easily be done by others who are interested. I think the important point is to acknowledge that the univariate effects are not all that there is, and this has now been integrated into the piece very well.

I thank the authors again for their constructive engagement, and I look forward to seeing the paper progress.